# Resmetirom Ameliorates NASH-Model Mice by Suppressing STAT3 and NF-κB Signaling Pathways in an RGS5-Dependent Manner

**DOI:** 10.3390/ijms24065843

**Published:** 2023-03-19

**Authors:** Xiaojing Wang, Liangjing Wang, Lin Geng, Naoki Tanaka, Bin Ye

**Affiliations:** 1Department of Gastroenterology, Lishui Hospital of Zhejiang University/The Central Hospital of Lishui, Lishui 323000, China; 2Department of Gastroenterology, The Second Affiliated Hospital, School of Medicine, Zhejiang University, Hangzhou 310009, China; 3Institute of Gastroenterology, Zhejiang University, Hangzhou 310009, China; 4Department of Global Medical Research Promotion, Shinshu University Graduate School of Medicine, Matsumoto 390-8621, Japan; 5International Relations Office, Shinshu University School of Medicine, Matsumoto 390-8621, Japan; 6Research Center for Social Systems, Shinshu University, Matsumoto 390-8621, Japan

**Keywords:** NASH, resmetirom, RGS5, STAT3, NF-κB

## Abstract

Resmetirom, a liver-directed, orally active agonist of THR-β, could play a favorable role in treating NASH, but little is known about the underlying mechanism. A NASH cell model was established to test the preventive effect of resmetirom on this disease in vitro. RNA-seq was used for screening, and rescue experiments were performed to validate the target gene of the drug. A NASH mouse model was used to further elucidate the role and the underlying mechanism of resmetirom. Resmetirom effectively eliminated lipid accumulation and decreased triglyceride (TG) levels. In addition, repressed RGS5 in the NASH model could be recovered by resmetirom treatment. The silencing of RGS5 effectively impaired the role of resmetirom. In the NASH mouse model, obvious gray hepatization, liver fibrosis and inflammation, and increased macrophage infiltration were observed in liver tissues, while resmetirom almost returned them to normal conditions as observed in the control group. Pathological experimental data also confirmed that resmetirom has great potential in NASH treatment. Finally, RGS5 expression was suppressed in the NASH mouse model, but it was upregulated by resmetirom treatment, while the STAT3 and NF-κB signaling pathways were activated in NASH but inhibited by the agent. Resmetirom could improve NASH by recovering RGS5 expression and subsequently inactivating the STAT3 and NF-κB signaling pathways.

## 1. Introduction

In the past four decades, nonalcoholic fatty liver disease (NAFLD) has become the foremost reason for cirrhosis and hepatocellular carcinoma, with a global prevalence of around 25% of the adult population, and its incidence has gradually increased every year [1,2,3]. Nonalcoholic steatohepatitis (NASH) is the inflammatory subtype of NAFLD and is characterized by necro-inflammation and faster fibrosis progression [4]. Unlike steatosis alone, which is not correlated with an increase in liver-related morbidity or mortality, NASH can further develop into more serious stages, including cirrhosis and hepatocellular carcinoma, and eventually result in liver failure and liver transplantation [5].

The mechanisms of the development and progression of NAFLD are extremely complex and involve many factors. Currently, the two-hit theory is one of the most mature hypotheses that can relatively fully summarize the mechanisms underlying the development of NAFLD. The first hit is hepatocyte steatosis induced by a high-fat diet (HFD), obesity, and insulin resistance, and the second hit is the inflammatory response induced by hepatic steatosis via oxidative stress and lipid peroxidation, and all these processes can result in fatty liver disease [6,7]. Due to the complicated pathophysiology and substantial heterogeneity of disease phenotypes, NASH is under-recognized in clinical practice, and there are no NASH-specific therapies currently approved by the US Food and Drug Administration. In fact, as NASH is strongly related to obesity, dyslipidemia, type 2 diabetes, and metabolic syndrome, a healthy lifestyle and weight control are still believed to be crucial for the prevention and treatment of NASH [8,9].

RGS5 is a member of the regulators of G protein signaling (RGS) family and found to take part in inflammation regulation [10]. A recent study reported that RGS5 could impair the progression of NAFLD by preventing the hyperphosphorylation of transforming growth factor beta-activated kinase 1 (TAK1) and the activation of the downstream JNK/p38 signaling cascade [11]. This finding makes RGS5 a potential target for NASH treatment. On the other hand, multiple signaling pathways are involved in the progression of NAFLD/NASH, such as JAK/STAT3 [12,13], NF-κB [14], and Toll-like receptor [15] signaling pathways. As expected, all these pathways play important roles in immune regulation.

Although no pharmaceuticals are currently approved for NASH, some drugs, such as vitamin E (an antioxidant) and pioglitazone (a thiazolidinedione insulin sensitizer, acting as a PPAR-γ agonist), have shown some benefit in randomized trials [16]. Currently, there is an urgent need to identify more efficient and low-toxicity drugs for NASH treatment. The thyroid hormone (TH) regulates glucose and lipid metabolism in the liver and also controls energy homeostasis [17]. It has been reported that thyroid hormone receptor β (THR-β) is abundantly expressed in hepatocytes and plays an important role in regulating the metabolic pathways frequently impaired in NAFLD/NASH in the liver [18]. In a phase 2 trial, resmetirom, an agonist for THR-β, resulted in a significant reduction in hepatic fat after 12 weeks and 36 weeks of treatment in patients with NASH [19]. Recently, another clinical trial reported that resmetirom can reduce hepatic fat in NASH patients and significantly improve their quality of life [20].

Although resmetirom shows significant potential in NASH treatment, the underlying molecular mechanisms still need to be further elucidated. In this study, a NASH cell model and a mouse model were established to confirm the intervention functions of resmetirom. mRNA sequencing and bioinformatic analysis were used to screen for the target gene of resmetirom, and a rescue experiment was performed to further validate its role in the protective functions of the agent in NASH. Our work provides a novel insight into the underlying mechanisms of resmetirom in NASH treatment.

## 2. Results

### 2.1. Resmetirom Effectively Impaired Lipid Accumulation in Liver Cells Induced by Oleic Acid

A NASH cell model was generated using oleic acid to elucidate the roles of resmetirom, as discussed in a previous publication [21]. In this study, both human and mouse NAFLD/NASH cell models were established using HepG2 cells and NCTC 1469 cells, respectively. As shown in Figure 1A, both cell lines were incubated with the indicated concentrations of oleic acid for 48 h, and the data from the Oil red O staining assay revealed that 1.2 mM oleic acid significantly elevated lipid build-up. Next, intracellular TG (triglyceride) levels were also examined in the two NAFLD/NASH cell models. As shown in Figure 1B, ELISA assays were performed on HepG2 and NCTC 1469 cells treated with 1.2 mM oleic acid for 48 h, and the data showed that the TG levels were significantly upregulated in the NAFLD/NASH cell models compared with those in the control. Before we tested the intervention effect of resmetirom on NAFLD/NASH, its cytotoxicity was tested in the two cell lines by using the MTT assay. As shown in Figure 1C, 48 h treatment with resmetirom markedly inhibited cell viability in a dose-dependent manner, and 200 μM of the drug significantly suppressed cell survival by about 50%. Next, 0 μM, 50 μM, 100 μM, and 200 μM resmetirom were chosen to test its effect on the NAFLD/NASH cell models. As shown in Figure 1D, pre-treatment with resmetirom markedly decreased lipid accumulation in both cell lines. Although the effects of 200 μM resmetirom were slightly better than those of 100 μM of the drug in HepG2 and NCTC 1469 cells, 100 μM resmetirom was used in subsequent investigations due to its negligible toxicity. Finally, the results of ELISA assays also confirmed that resmetirom (100 μM and 200 μM) effectively improved NAFLD/NASH, as TG levels were downregulated by the drug treatment (Figure 1E).

### 2.2. mRNA Sequencing Indicated That Resmetirom Might Improve NAFLD/NASH by Influencing the Activation of Toll-like Receptor and Jak-STAT3 Signaling Pathways

In the present study, mRNA sequencing was performed to compare the mRNA expression profiles in NAFLD/NASH cells and in NAFLD/NASH cells treated with resmetirom. First, three replicate cell samples were prepared from HepG2 cells, and Oil red staining experiments were used to validate NAFLD/NASH establishment and the intervention efficiency of resmetirom; the obtained data are shown in Figure 2A. Three parallel samples were then subjected to mRNA sequencing, and volcano plots (Figure 2B) as well as heat maps (Figure 2C) were produced using the DEGs obtained from the comparisons of NASH vs. control and NASH_R vs. NASH. More importantly, GO and KEGG enrichment analyses were also performed on the DEGs. As shown in Figure 2D, GO enrichment analysis revealed that the extracellular region, space, and matrix were enriched in the two group comparisons, while insulin-like growth factor binding was enriched in the comparison of NASH_R vs. NASH. In the KEGG enrichment analysis, ECM–receptor interaction, cytokine–cytokine receptor interaction, the TGF-beta signaling pathway, and the PI3K/AKT pathway were enriched in the comparison of NASH vs. control (Figure 2E, upper panel). Several important signaling pathways, such as the Toll-like receptor and Jak-STAT3, were enriched in the comparison of NASH_R vs. NASH (Figure 2E, lower panel). All these findings suggest that resmetirom might improve NAFLD/NASH by influencing the activation of the Toll-like receptor and Jak-STAT3 signaling pathways, as both of them have been proven to take part in regulating inflammation and contribute to the development of NASH [22,23].

### 2.3. Resmetirom Might Ameliorate NAFLD/NASH in an RGS5-Dependent Manner

We next screened for the target gene by which resmetirom could improve NAFLD/NASH in the HepG2 cell model. First, cross-enrichment analysis was performed in the DEGs from the two comparisons of NASH vs. control and NASH_R vs. NASH. Among the DEGs, only five genes, *CHAC1*, *SLC7A11*, *COL4A3*, *EGR3*, and *RGS5*, which were influenced in NASH but recovered by resmetirom treatment, were further validated by qPCR in the RNA samples used in sequencing. As shown in Figure 3A, all five genes except CHAC1 could be recovered by resmetirom treatment to some extent, but significance was only observed in the expression of *RGS5*. This finding suggests that RGS5 might be a potential target gene for resmetirom in the NAFLD/NASH cell model. Therefore, we first tested the protein expression of RGS5 in HepG2 cells incubated with different concentrations of oleic acid for 48 h, and the data from a Western blot assay indicated that RGS5 protein expression was markedly decreased by oleic acid treatment in a time-dependent manner (Figure 3B). Meanwhile, the protein expression of several NAFLD/NASH-related genes, including endoplasmic-reticulum-stress-related PERK and IRE-1α [24], as well as the phosphorylation of STAT3 and NF-κB (p65), was examined in the resmetirom-treated NAFLD/NASH model established with HepG2 cells. As shown in Figure 3C, RGS5 protein expression was suppressed in NAFLD/NASH cells but was recovered effectively by resmetirom treatment, and the phosphorylation of both STAT3 and NF-κB, which were activated in NAFLD/NASH, was significantly attenuated by resmetirom treatment. However, the expression of PERK and IRE-1α was not notably influenced in either the untreated NAFLD/NASH model or the resmetirom-treated model.

To further confirm whether resmetirom improved NAFLD/NASH by upregulating RGS5 expression, siRNA against RGS5 was used in this study. First, HepG2 cells were transfected with three siRNAs against *RGS5* or negative control siRNA (NC) for 24 h, and qPCR and Western blot assays were performed to determine the silencing efficiency of siRNA. As shown in Figure 3D,E, all three siRNAs could significantly suppress the expression of *RGS5* mRNA and proteins in HepG2 cells, and *RGS5* siRNA-2 had a very high silencing efficiency. Therefore, we next tested whether *RGS5* silencing could attenuate the role of resmetirom in NAFLD/NASH. As shown in Figure 3F, resmetirom treatment effectively decreased lipid accumulation in the NAFLD/NASH cell model, while RGS5 knockdown markedly impaired the role of resmetirom. Subsequently, the silencing of RGS5 also suppressed the functions of resmetirom in the activation of STAT3 and NF-κB signaling pathways in the NAFLD/NASH cell model (Figure 3G). These findings support the idea that resmetirom might play its role in NAFLD/NASH by upregulating RGS5 expression.

### 2.4. Resmetirom Significantly Prevented NASH in Mouse Model

The NASH mouse model was generated in this study as described in a previous study [25]. C57BL/6J mice (male, 6 weeks old) were fed a normal diet as a control or an AMLN diet to generate the NASH mouse model. The body weights of the mice in the two groups increased slowly as time went on, but those in the AMLN group increased more quickly than the control. After twenty-five weeks of feeding, the average body weight of mice in the AMLN group reached about two times that in the control group (Figure 4A). We took one mouse from each group to sacrifice to observe the status of the liver, and as shown in Figure 4B, the liver from the AMLN group was markedly enlarged compared with that from the control group. Moreover, the liver from the AMLN group was visibly whitish, probably due to substantial lipid accumulation (Figure 4B). ELISA experiments were applied to determine HDL-C, LDL-C, TG, and T-chol in serum. As shown in Figure 4C, the serous levels of both HDL-C and LDL-C were significantly elevated in the AMLN group, while the T-chol level had slightly increased with no significance. However, and interestingly, the levels of TG decreased after AMLN diet feeding, although no significance was observed (Figure 4C).

Pathological experiments were performed to confirm the pathological changes in the livers. As shown in Figure 4D, the data from the Oil red O staining assay revealed that extreme lipid accumulation was observed in the liver from the AMLN group, which also confirmed the reason for the gray hepatization. On the other hand, a Masson assay was used to detect fibrosis in the livers, and our data showed that AMLN diet feeding resulted in serious fibrosis in the liver (Figure 4E). HE staining also was performed to identify differences in the pathological characteristics of livers between the normal and NASH-group mice, and apparent voids were observed in the livers of NASH mice due to the presence of large oil droplets, which is also called macrovesicular steatosis (Figure 4F). In addition to this, the infiltration of inflammatory cells such as macrophages around steatotic hepatocytes, with a so-called crown-like appearance, was also found in liver tissue from the NASH mice, which was supported by the increased number of F4/80-positive cells (Figure 4G). As fibrosis and inflammation are the most classic characteristics of NASH, the expression levels of a number of related genes were also examined in this study. As shown in Figure 4H, significantly higher levels of the mRNAs of *collagen 1a1* (*Col1a1*), key drivers of fibrogenesis such as connective tissue growth factor and galectin 3 (encoded by *Ctgf* and *Lgals3*, respectively), and SPP1 were observed in NASH livers. In addition to increased fibrosis, the expression of mRNA-encoding macrophage marker CD68 and typical pro-inflammatory cytokines/chemokines, such as tumor necrosis factor-α (TNFα, encoded by *Tnf*), interleukin (IL)-1β (*Il1b*), chemokine (C-C motif) ligand 2 (*Ccl2*), and colony-stimulating factor 1 (*Csf1*), was significantly increased in NASH livers compared with the normal control (Figure 4I).

We next took sixteen mice from the AMLN group for resmetirom treatment; eight of these mice were treated with a dose of 3 mg/kg, and the other eight mice were treated with a dose of 5 mg/kg daily, while the remaining mice were fed under the same conditions as previously used. The body weight of all mice was measured every four days, and the two different doses of resmetirom effectively decreased the body weight compared with the mice in the NASH group (Figure 5A). After forty-eight days of feeding, gray hepatization was effectively improved by resmetirom treatment, especially at the higher dose (5 mg/kg, Figure 5B,C). Finally, the ratio of liver to body weight was also calculated, and as shown in Figure 5D, the weight ratio was significantly upregulated in NASH-model mice, while it was markedly attenuated by resmetirom treatment at both of the two doses. Our data also indicated that F4/80-positive cell counts were effectively decreased in NASH livers treated with resmetirom, which implies that macrophage infiltration was reversed by the drug treatment (Figure 5E). More importantly, the mRNA expression of all inflammation- and liver-fibrosis-related genes were markedly downregulated by resmetirom treatment in NASH livers compared with the control (Figure 5F,G).

### 2.5. Resmetirom Effectively Improved NASH In Vivo by Recovering RGS5 Expression and Inactivating STAT3 Signaling Pathway

Abnormal lipid accumulation in the liver is the most important marker for NASH; therefore, an Oil red staining assay was performed in sections of livers from the different groups: control, NASH, low-dose resmetirom treatment, and high-dose resmetirom treatment. As shown in Figure 6A, serious lipid accumulation was observed in liver tissues from the NASH group, while resmetirom treatment effectively reduced this condition, especially in the group that received high-dose resmetirom treatment. The data from HE staining also identified that high-dose resmetirom treatment almost reversed the pathological characteristics of the liver to the status of a normal liver (Figure 6B). We also tried to test cell apoptosis in the livers of the NASH mouse model and the resmetirom-treated mice. However, there was no obvious cell apoptosis observed in the livers of NASH mice, which might be attributed to the early stage of NASH we established in this study (Figure 6C). Finally, a Masson staining experiment was performed to observe fibrosis in the liver. Our data indicated that resmetirom treatment effectively improved the fibrosis state induced by NASH, and the high dose (5 mg/kg) of the agent exerted a better effect (Figure 6D).

Our data from the cell experiments implied that RGS5 might play a critical role in the effect of resmetirom on NASH. Therefore, we next examined RGS5 expression in liver tissues using an IHC assay. As shown in Figure 7A, the RGS5 protein level was prominently decreased in the liver in the NASH group, and resmetirom treatment effectively restored the expression of RGS5. Moreover, Western blot assays were used to determine RGS5 expression and the activation of STAT3 and NF-κB signaling pathways. As shown in Figure 7B, similar expression profiles of RGS5 were observed in the data from the WB analysis, and STAT3 and NF-κB phosphorylation were also markedly upregulated in NASH livers and significantly attenuated by resmetirom.

## 3. Discussion

It is well known that NAFLD/NASH is strongly associated with obesity, dyslipidemia, type 2 diabetes, and metabolic syndrome, but the detailed pathogenic mechanism is very complicated and has not been well elucidated [26]. Unfortunately, no NASH-specific therapies are currently approved by the Food and Drug Administration of China or the US, despite the mortality rate of patients with NASH being substantially high and growing steadily [27]. Therefore, efforts to both evaluate the pathogenic mechanism and develop medicine for NAFLD/NASH are in urgent need.

The signaling pathways downstream of thyroid hormone receptors (THRs) play fundamental roles in regulating organogenesis, growth, and differentiation and significantly influence energy metabolism, lipid utilization, and glucose homeostasis, implying the potential value of liver disease treatment by targeting THRs [28]. Although little benefit was obtained from thyromimetics in improving NASH due to unwanted side effects, significant therapeutic effects on NASH were observed with resmetirom (MGL-3196), a new oral, liver-directed, highly selective thyroid receptor β agonist [19,29,30]. However, there has been little attention paid to the need to evaluate the underlying mechanism by which resmetirom exerts its intervention effect on NASH until now. In the present study, we have tried to clarify how resmetirom functions in treating NASH.

We first established a cell model for NAFLD/NASH in human hepatoma cell line HepG2 cells and mouse liver cell line NCTC 1469 cells using oleic acid, as described in previous publications [31,32]. The establishment of NAFLD/NASH models was validated by the substantial lipid accumulation via an Oil red O staining assay and by elevated intracellular TG levels via an ELISA assay. In addition to a high dose of 200 μM resmetirom, a low dose of 100 μM resmetirom with 10% cell viability inhibition was also used to test its role in NAFLD/NASH in order to eliminate the potential influence of cytotoxicity. Our data indicated that oleic acid treatment markedly induced lipid accumulation in both liver cell lines, while 100 μM resmetirom effectively restored lipid accumulation and TG levels to similar levels to those of the control cells. These findings are consistent with the role of thyroid receptor β in lipid metabolism in the liver reported previously [33]. Although a slightly better effect was observed in the group with 200 μM agent treatment, the concentration of resmetirom in further experiments should be 100 μM due to the significant cytotoxicity of the high dose of the drug in both cell lines. In animal experiments, 5 mg/kg resmetirom effectively restored the liver appearance of NASH-model mice to the normal level in the control group and restored the ratio of liver weight to body weight, which was higher in NASH models. The data from pathological analysis also showed that resmetirom successfully improved NASH, which was supported by the eliminated lipid accumulation, the normal pathological characteristics of the liver, decreased macrophage infiltration, and the downregulation of the mRNA expression of genes associated with liver fibrosis and inflammation. As mentioned above, the pathophysiology of NASH is extremely complicated, but it is well known that liver-resident macrophages and recruited macrophages play a central part in the progression of the disease [34,35]. Our data imply that the role of resmetirom in NASH might involve macrophages.

To the best of our knowledge, there is little published research focused on the molecular mechanisms of resmetirom in NASH treatment. Therefore, mRNA sequencing was conducted to confirm the gene expression profiling differences between NASH and the control, as well as between resmetirom-treated NASH and NASH alone. Hundreds of differently expressed genes were obtained from the two group comparisons, and GO and KEGG enrichment analyses were also performed. Several NASH-related signaling pathways were enriched, such as ECM–receptor interaction, cytokine–cytokine receptor interaction, and the TGF-beta signaling pathway, as well as the PI3K/AKT signaling pathway, in the comparison of NASH vs. control; in addition, the Toll-like receptor and Jak-STAT3 were enriched in the comparison of NASH_R vs. NASH. The NF-κB pathway is downstream of the PI3K/AKT signaling pathway, acting as a master regulator of inflammatory responses, and its abnormal activation is a key pathogenic factor for the development of NAFLD/NASH [36,37]. The activation of STAT3 can play anti- or pro-inflammatory roles and take part in the pathogenesis of liver fibrosis in NAFLD/NASH [23,38]. Our data indicated that both NF-κB and Jak-STAT3 signaling pathways were activated in the NAFLD/NASH cell model while being inhibited by resmetirom treatment. Similar results were observed in the animal experiments. These findings suggest that resmetirom can improve NASH by inactivating the NF-κB and Jak-STAT3 signaling pathways, at least partially.

We also screened five potential target genes for resmetirom from DEGs obtained by sequencing; all these genes’ expression was regulated in the NAFLD/NASH cell model but was reversed by the drug treatment. However, only RGS5 was chosen for further investigation as the target gene of resmetirom due to the results from the qPCR assay. This was also supported by data from the Western blot assay, which showed that the protein levels of RGS5 were significantly downregulated in the NAFLD/NASH cell model. RGS5 is a member of the RGS family, which is involved in the regulation of heterotrimeric G proteins by acting as GTPase activators [39]. Several reports have indicated that RGS5 may also be a regulator of the inflammatory response. The downregulation of RGS5 promoted inflammation in endothelial cells and impaired their normal function in contributing to coronary artery disease [40]. More interestingly, a recent study reported that RGS5 is an essential regulator protecting against the progression of NAFLD by preventing its hyperphosphorylation of TAK1 and the activation of the downstream JNK/p38 signaling cascade [11]. This is the first and only report that identifies RGS5 as a promising target for the treatment of NAFLD. In this study, our data confirmed that RGS5 expression was significantly suppressed in both cell and animal NAFLD/NASH models. The silence of RGS5 using siRNA effectively attenuated the role of resmetirom in lipid accumulation in the NAFLD/NASH cell model and also restored the activation of the NF-κB and Jak-STAT3 signaling pathways inhibited by the drug treatment. However, more experiments in transgenic mice with RGS5 knockout should be performed in the future to further verify whether RGS5 is indispensable for the function of resmetirom in NASH treatment at the animal level.

## 4. Materials and Methods

### 4.1. Cell Culture

Human hepatoma cell line HepG2, used as human normal liver cells, was obtained from ATCC (American Type Culture Collection, Rockville, MD, USA), and the mouse normal liver cell line NCTC 1469 was purchased from Procell Bio (Wuhan, China). The two cell lines were cultured in RPMI 1640 medium supplemented with 10% FBS (fetal bovine serum), 1% P/S (penicillin and streptomycin), and 2 mM L-glutamine in a humidified incubator with 5% CO_2_ at 37 °C.

### 4.2. In Vitro Cell Culture Model for NAFLD/NASH Disease

In the present study, HepG2 and NCTC 1469 cells were used to construct NAFLD/NASH cell models by using oleic acid (Sigma, St. Louis, MO, USA) as described previously [41]. In brief, the two cell lines were incubated with different concentrations of oleic acid (0 mM, 0.6 mM, 0.9 mM, and 1.2 mM) for 48 h. After treatment, Oil red O staining was performed using a kit from Beyotime Bio (Shanghai, China) according to the manufacturer’s instructions.

### 4.3. MTT Assay for Resmetirom Cytotoxicity Detection

HepG2 and NCTC 1469 cells were seeded into each well of a 96-well plate at densities of 5000 or 7000 cells, respectively, and cultured overnight to allow cell attachment. Then, cells were treated with different doses of resmetirom (0 mM, 100 mM, 200 mM, 300 mM, 400 mM, and 500 mM, obtained from Selleck) for 48 h. After that, the medium was removed, and 50 μL of MTT solution (1mg/mL in PBS) was added to each well to incubate for 2–4 h at 37 °C. Subsequently, 150 DMSO was added to each well, and the solution was shaken to dissolve the purple crystals completely. Finally, the absorbance at 570 nM was read using a microplate reader (Molecular Device).

### 4.4. mRNA Sequencing and Bioinformatic Analysis

An mRNA-sequencing experiment was performed on HepG2 cells to further elucidate the underlying mechanisms by which resmetirom can improve NAFLD in vitro. HepG2 cells were pre-incubated with 0 or 150 μM resmetirom for 48 h and then treated with 0- or 1.2-mM oleic acid for another 48 h to generate the NAFLD cell model. After that, total RNA was extracted using Trizol reagent (Invitrogen, Inc., Carlsbad, CA, USA), according to the manufacturer’s protocol, from samples from three dependent experiments. The mRNA-sequencing assay and the bioinformatic analysis, including GO and KEGG enrichment, were performed using LC-Bio (Hanghzou, China).

### 4.5. qRT-PCR Assay

For further validation of the expression profiling of the differentially expressed genes obtained from mRNA sequencing, a qRT-PCR experiment was performed. In brief, the cDNA (complementary DNA) was generated by using M-MLV reverse transcriptase (Promega, Fitchburg, WI, USA), and the qPCR experiment was carried out on an ABI-7500 thermal cycler using a matched real-time PCR master mixture kit (Thermofisher, Waltham, MA, USA). The relative mRNA expression was determined using the relative standard curve method (2^−ΔΔCt^), for which β-actin was used as the reference. The primers used in this study are listed in Table 1.

### 4.6. Western Blot Assay

Proteins were obtained from both cell samples and animal tissue samples by using RIPA lysis buffer containing a 1% protease inhibitor cocktail (Sigma, St. Louis, MO, USA). For protein extraction from animal samples, a tissuelyser was employed to improve the extraction efficiency. The protein concentration for each sample was determined using a BCA assay kit (Thermofisher, Waltham, MA, USA), and 20 μg of protein from each sample was loaded and separated on 8%, 10%, or 15% SDS–PAGE gels by electrophoresis at 200 V for 1 h. Then, proteins were transferred from the gel to a PVDF membrane (0.45 μm, Millipore, Bedford, MA, USA) and blocked with 5% fat-free milk in PBS for 2 h at room temperature. After that, the membranes were incubated with the indicated primary antibodies overnight at 4 °C, followed by another 2 h incubation with the corresponding horseradish peroxidase-conjugated secondary antibodies. Finally, signals were visualized using an ECL kit (Thermofisher, Waltham, MA, USA) and the band was quantified using Image J software1.51j8 (National Institute of Health, Bethesda, MA, USA). β-Actin was used as a loading control.

The details of the antibodies used in this study were as follows: RGS5 (1:1000, Proteintech, 11590-1-AP), PERK (1:1000, Proteintech, 24390-1-AP), IRE-1α (1:1000, CST, 3294), p-STAT3 (1:1000, CST, 9145), STAT3 (1:1000, CST, 4904), NF-κB (1:1000, CST, 8242), p-NF-κB (1:1000, CST, 3033), and β-actin (1:5000, CST, 4970).

### 4.7. siRNA and Transfection

To test whether RGS5 plays an indispensable role in the effect of resmetirom on NAFLD/NASH, RGS5 siRNA was used to perform rescue experiments. Three siRNAs against RGS5 and one NC siRNA were obtained from GemmaPharma-Bio (Shanghai, China). The sequences were as follows: NC siRNA, sense: 5′-UUCUCCGAACGAG UCACGUTT-3′, antisense: 5′-ACGUGACUCGUUCGGAGA ATT-3′; RGS5 siRNA-1, sense: 5′-CCUCCAUAAUAACCCUGCAUUTT-3′, antisense: 5′-AAUGCAGGGUUAUUAUGGAGGTT-3′; RGS5 siRNA-2, sense: 5′-AGAUGGCUGAGAAGGCAAATT-3′, antisense: 5′-UUUGCCUUCUCAGCC AUCUTT-3′; RGS5 siRNA-3, sense: 5′-GCGUGAUUCCCUGGACAAATT-3′, antisense: 5′-UUUGUCCAGGG AAUCACGCTT-3′. For transfection, lipofectamine 2000 reagent was used in this study. In brief, siRNA was incubated with the transfection reagent for 20 min to allow the formation of the complex, and then the siRNA–lipo complex was added to the cell culture very gently, drop-by-drop. After 24 h incubation, proteins were extracted and the silencing efficiency was determined using a Western blot assay.

### 4.8. Animal Model for NASH Disease and Resmetirom Treatment

Twenty C57BL/6J mice (male, 6 weeks old) were purchased from Vital River (Beijing, China) and allowed to acclimate for 1 week. Then, four mice were fed a normal diet as a control, and the other mice were fed the Amylin Liver NASH (AMLN) diet to generate the NASH mouse model as described previously [42]. In the process of model establishment, the body weights of the mice from the two groups were measured every eight days, and blood from the caudal vein was obtained for the detection of HDL-C, LDL-C, TG, and T-chol by ELISA assays after an obvious difference in body weight between the two groups of mice was observed.

After finishing the model establishment, ten mice in the AMLN diet group were divided into two groups (either 3 mg/kg resmetirom treatment or 5 mg/kg resmetirom treatment, eight mice for each group), and all NASH mice were fed a fixed amount of the AMLN diet (5 g/day/mouse). The body weight was measured for all mice every four days.

### 4.9. ELISA

ELISA kits for the detection of HDL-C, LDL-C, TG, and T-chol for both cell samples and serum samples were obtained from MEIMIAN-Bio (Yancheng, China). Experiments were performed according to the manufacturer’s instructions, and all data were obtained from triplicated experiments.

### 4.10. Pathological Experiment

#### 4.10.1. Oil Red O Staining

Liver tissues were freshly frozen, cut into sections (5 μm thick), and mounted on slides. The sections were air-dried for 30 min at room temperature and then fixed in 10% ice-cold formalin for 10 min, followed by air drying again for another 30 min. After being rinsed with 60% isopropanol, the sections were stained with a freshly prepared Oil red O working solution for 15 min and rinsed again with 60% isopropanol. After five quick dips in alum hematoxylin to stain the nuclei and rinsing with distilled water, the sections were then mounted in glycerin jelly.

#### 4.10.2. Masson Staining

Liver tissues were fixed with 10% formalin, paraffin-embedded, cut into sections (5 μm thick), and mounted on slides. The sections were then deparaffinized and rehydrated with 100% alcohol, 95% alcohol, and 70% alcohol and rinsed with distilled water. They were then stained in Weigert’s iron hematoxylin working solution for 10 min, rinsed in warm running tap water for 10 min, and washed in distilled water three times. Then, they were stained in Biebrich scarlet-acid fuchsin solution for 10–15 min and rinsed with distilled water. The sections were then differentiated in a phosphomolybdic-phosphotungstic acid solution for 10–15 min, directly transferred to an aniline blue solution, and stained for 5–10 min. They were then rinsed briefly in distilled water and differentiated in a 1% acetic acid solution for 2–5 min. After washing with distilled water again, they were dehydrated very quickly with 95% ethyl alcohol and absolute ethyl alcohol and cleared in xylene. Then, the sections were mounted with a resinous mounting medium.

#### 4.10.3. HE Staining

Liver tissues were fixed with 10% formalin, paraffin-embedded, cut into sections (5 μm thick), and mounted on slides. The sections were then deparaffinized and rehydrated with 100% alcohol, 95% alcohol, and 70% alcohol and rinsed with distilled water. Sections were then subjected to Hematoxylin and Eosin staining (H&E staining). In brief, the nuclei were stained with the alum hematoxylin first and rinsed in running tap water. They were differentiated with 0.3% acid alcohol and then rinsed in running tap water. Then, the sections were rinsed in Scott’s tap water substitute and rinsed in tap water, followed by staining with eosin for 2 min. Finally, after dehydration and clearing, they were mounted with a resinous mounting medium.

#### 4.10.4. TUNEL Assay

Liver tissues were fixed with 10% formalin, paraffin-embedded, cut into sections (5 μm thick), and mounted on slides. To assess cell death in liver tissues, TUNEL was performed in this study. A commercially available TUNEL assay kit was obtained from Abcam, and the experiment was performed according to the manufacturer’s instructions. The nuclei were stained with alum hematoxylin.

#### 4.10.5. IHC Assay

In this study, RGS5 expression in the liver was examined by using an IHC assay. For the IHC assay, liver tissues were fixed with 10% formalin, paraffin-embedded, cut into sections (5 μm thick), and mounted on slides. After deparaffinization, rehydration, and antigen retrieval, the sections were incubated with primary antibodies (F4/80 and RGS5, both diluted at 1:100 in BSA) at room temperature for 2 h. After washing with distilled water three times, the sections were incubated with horseradish peroxidase-conjugated goat anti-rabbit IgG. Slides were incubated with diaminobenzidine substrate for color development, and the nuclei were stained with alum hematoxylin.

### 4.11. Statistical Analysis

All data are expressed as means ± SD (for data from cell experiments) or means ± SEM (for data from animal experiments). The two-tailed unpaired Student’s *t*-test was used for the statistical analysis of differences between the two groups, and one-way ANOVA was used for multiple groups, using GraphPad Prism 5.0 (GraphPad Software, Inc., La Jolla, CA, USA); *p* < 0.05 (*) was considered significant.

## 5. Conclusions

For the first time, our study has documented the underlying mechanisms of resmetirom in improving NASH at both the cell and animal levels. Our data indicate that resmetirom can effectively ameliorate NASH by suppressing STAT3 and NF-κB signaling pathways in an RGS5-dependent manner (Figure 7D). These findings provide a novel insight into the mechanisms by which resmetirom exerts anti-NASH effects.

## Figures and Tables

**Figure 1 ijms-24-05843-f001:**
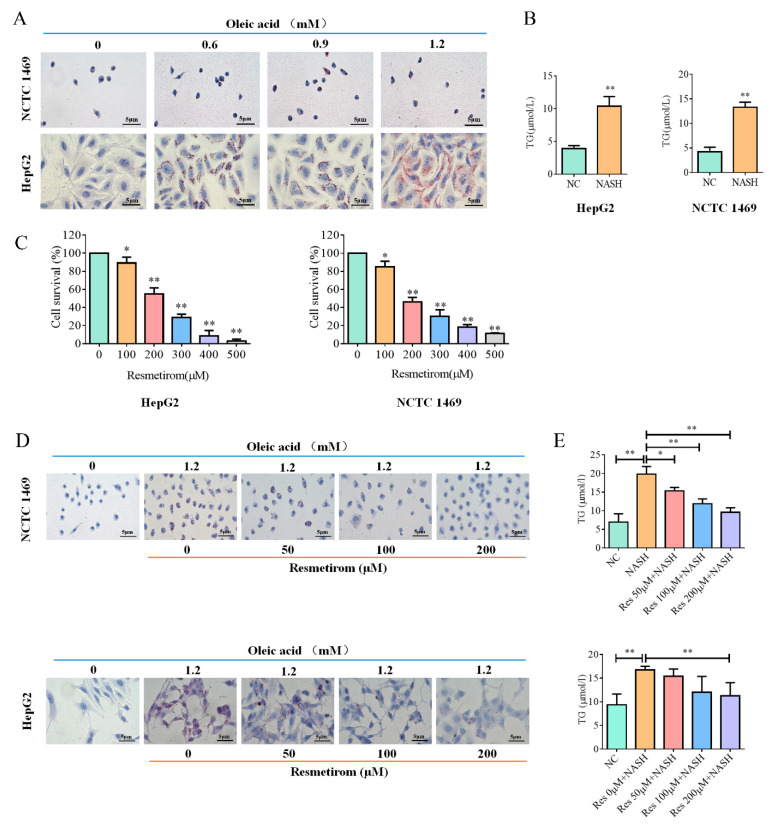
Resmetirom effectively improved NASH in cell models generated using oleic acid. (**A**), HepG2 and NCTC 1469 cells were treated with the indicated doses of oleic acid for 48 h, and Oil red O staining experiment was performed to determine the intracellular lipid accumulation. (**B**), ELISA assay was performed to detect the intracellular TG levels in cells treated as in (**A**). (**C**), HepG2 and NCTC 1469 cells were treated with the indicated concentrations of resmetirom for 48 h, MTT assay was used to assess the cytotoxicity of the drug. (**D**), HepG2 and NCTC 1469 cells were pre-incubated with the indicated concentrations of resmetirom for 48 h, followed by another 48 h treatment with the indicated doses of oleic acid to establish the NASH cell model, and Oil red O staining experiment was performed to determine the intracellular lipid accumulation. (**E**), ELISA assay was performed to detect the intracellular TG levels in cells treated as in (**D**). Data are presented by mean ± SD for three independent experiments. * *p* < 0.05; ** *p* < 0.01 vs. control.

**Figure 2 ijms-24-05843-f002:**
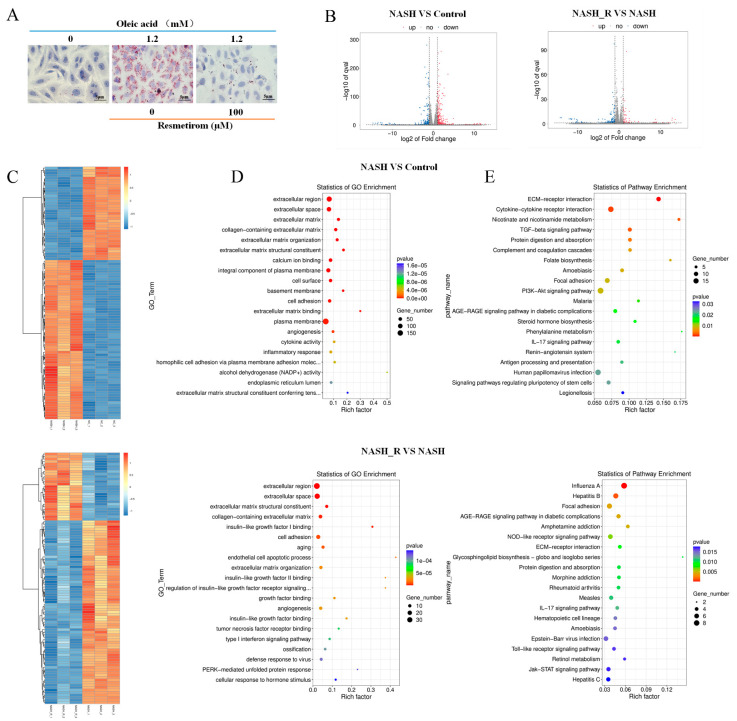
mRNA sequencing was performed to elucidate the mRNA expression profile in the resmetirom-treated NASH cell model. (**A**), HepG2 cells were pre-incubated with 0 or 100 μM resmetirom for 48 h, followed by treatment with 0- or 1.2-mM oleic acid for another 48 h, and Oil red O staining experiment was performed to determine the intervention effect of the drug. (**B**), Total RNA was extracted for mRNA-sequencing assay, differentially expressed genes (DEGs, fold change > 2, *p* < 0.05) were obtained, and volcano plots were generated using DEGs between NASH (NASH cell model) vs. control (left panel), and NASH_R (NASH cell model treated with 100 μM resmetirom) vs. NASH (right panel). (**C**), Heat maps were produced for DEGs between NASH vs. control (upper panel), and NASH_R vs. NASH (lower panel). (**D**), GO enrichment analysis was performed in DEGs between NASH vs. control (upper panel), and NASH_R vs. NASH (lower panel). (**E**), KEGG enrichment analysis was performed in DEGs between NASH vs. control (upper panel), and NASH_R vs. NASH (lower panel).

**Figure 3 ijms-24-05843-f003:**
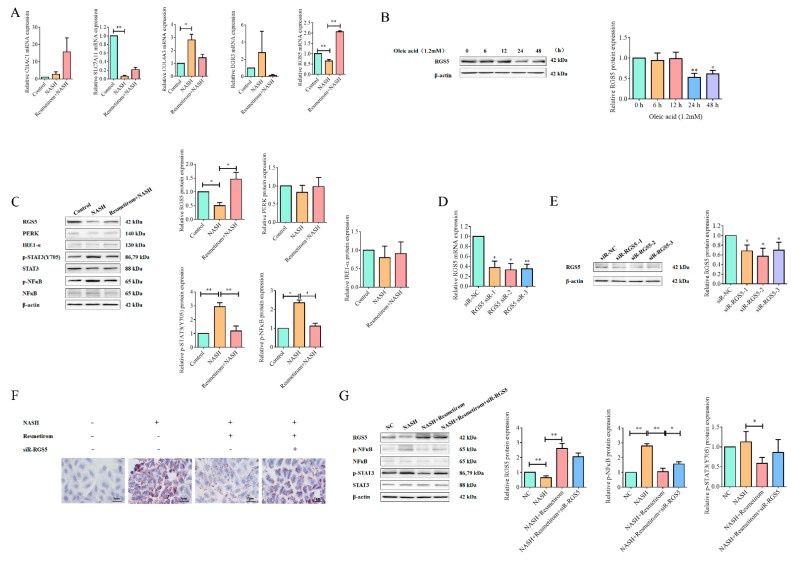
Resmetirom might ameliorate NASH in an RGS5-dependent manner. (**A**) The expressions of the top five DEGs were further validated by qPCR. (**B**), HepG2 cells were treated with indicated concentrations of oleic acid for 48 h, Western blot assay was used to determine the expression of RGS5 protein. The protein level was analyzed and is presented in the histogram in the right panel. (**C**), HepG2 cells were pre-incubated with 0 or 100 μM resmetirom for 48 h, followed by treatment with 0- or 1.2-mM oleic acid for another 48 h, and the indicated protein expressions were determined by Western blot assays. The protein level was analyzed and is presented in the histogram in the right panel. (**D**), HepG2 cells were transfected with RGS5 siRNA or NC siRNA for 24 h, and qPCR experiment was performed to examine the silence efficiency. (**E**), Total proteins were extracted in cells treated as in (**D**), and Western blot assay was performed to detect the RGS5 expression. The protein level was analyzed and is presented in the histogram in the right panel. (**F**), HepG2 cells were pre-incubated with 0 or 100 μM resmetirom for 48 h and transfected with RGS5 siRNA-2 or NC for 24 h, followed by treatment with 0- or 1.2-mM oleic acid for another 48 h, and Oil red O staining experiment was performed to determine the cellular lipid accumulation. (**G**), Total proteins were extracted in cells treated as in (**F**), and Western blot assay was performed to detect the expressions of the indicated proteins. The protein level was analyzed and is presented in the histogram in the right panel. Data are presented by mean ± SD for three independent experiments. * *p* < 0.05; ** *p* < 0.01 vs. control.

**Figure 4 ijms-24-05843-f004:**
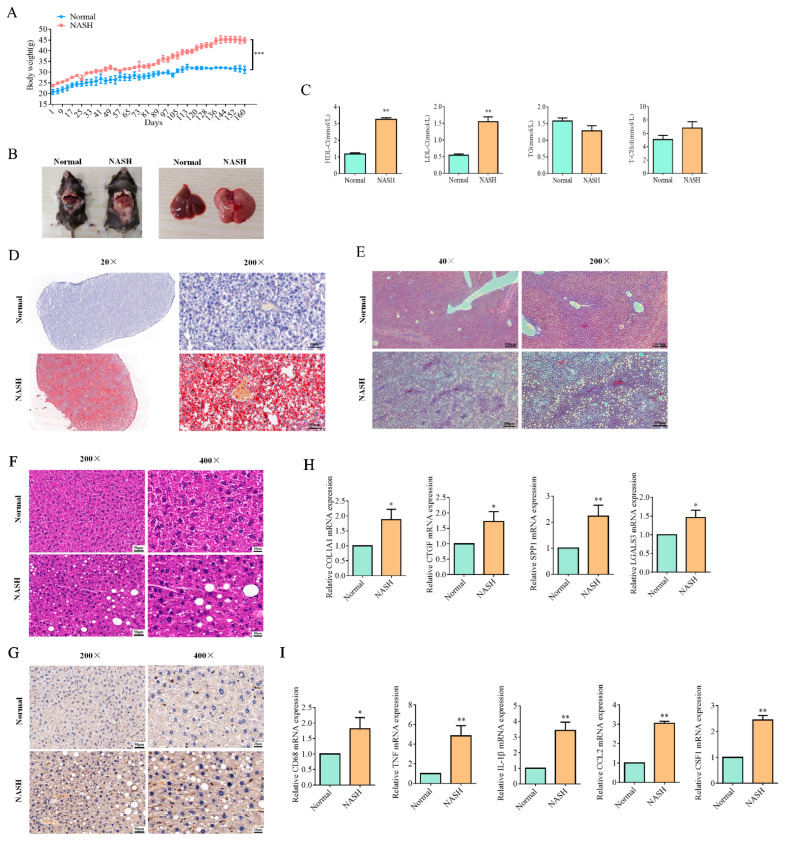
Establishment of NASH mouse models. (**A**), C57BL/6J mice (male, 6 weeks old) were fed with normal or AMLN diets, and body weights were measured every eight days. (**B**), After about 25 weeks feeding, one mouse from each group was sacrificed (left panel), livers were removed, and presented in right panel. (**C**), Serum was obtained from blood from caudal vein for detection of HDL-C, LDL-C, TG, and T-chol by ELISA assay. (**D**), Sections of livers from mice in B were used to perform the Oil red O staining assay to determine the lipid accumulation in liver tissue. (**E**), Sections of livers from mice in B were used in Masson staining experiment to confirm fibrosis status in liver tissues. (**F**), H&E staining was used to determine the pathological change in livers. (**G**), IHC assay was performed to confirm the pan-macrophage marker F4/80 in livers. (**H**), qPCR analysis of genes associated with liver fibrosis. (**I**), qPCR analysis of genes associated with inflammation. Data are presented by mean ± SEM for three independent experiments. * *p* < 0.05; ** *p* < 0.01 vs. control.

**Figure 5 ijms-24-05843-f005:**
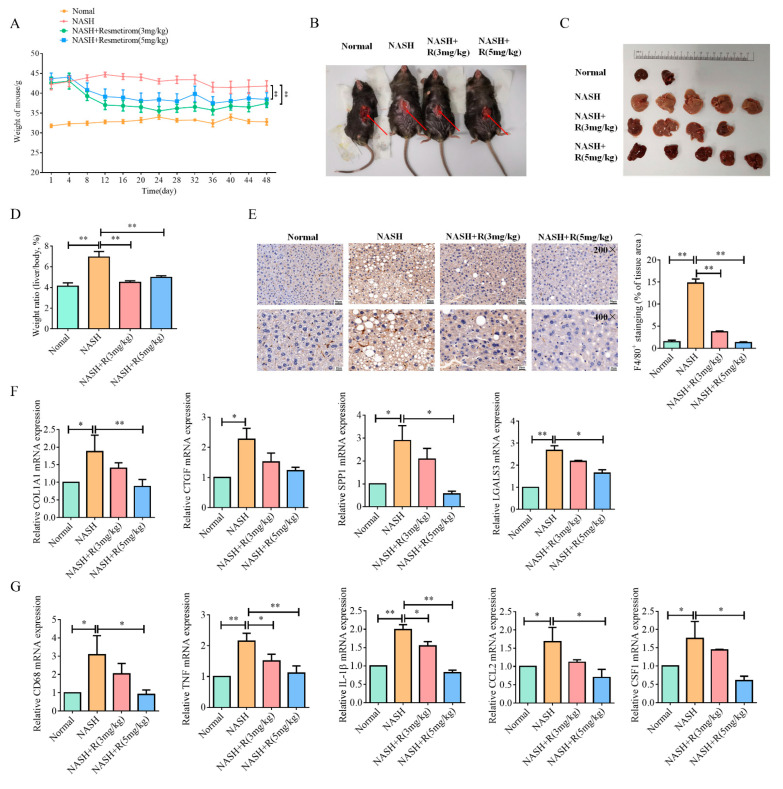
Resmetirom significantly suppressed NASH in mouse model generated by AMLN diet. After finishing the establishment of the NASH mouse models, all mice were continually feeding as previously, while sixteen mice of NASH group were treated with either a low dose (3 mg/kg, eight mice) or high dose (5 mg/kg, eight mice) of resmetirom every day. (**A**), Body weights were measured every four days. (**B**), After 48 days of treatment, mice were sacrificed and the presented mouse for each group is shown. The arrows indicate the positions of the liver. (**C**), Livers were removed and presented. (**D**), The ratios of livers to body weights were calculated and presented. (**E**), IHC assay was performed to confirm the pan-macrophage marker F4/80 in livers, and the infiltration of macrophages is quantified and statistically analyzed in right panel. (**F**), qPCR analysis of genes associated with liver fibrosis. (**G**), qPCR analysis of genes associated with inflammation. Data are presented by mean ± SEM for three independent experiments. * *p* < 0.05; ** *p* < 0.01 vs. control.

**Figure 6 ijms-24-05843-f006:**
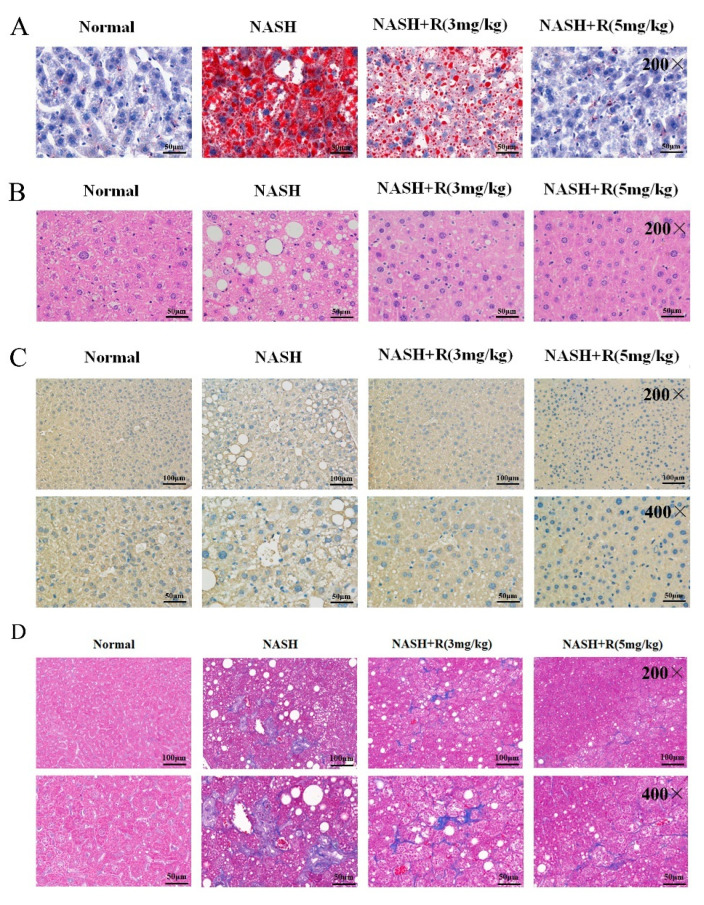
Resmetirom effectively inhibited lipid accumulation in NASH mouse model. (**A**), Sections of livers from mice in Figure 5 were used to perform the Oil red O staining assay to determine the lipid accumulation in liver tissue. (**B**), HE staining assay was used to confirm the pathological change in livers. (**C**), TUNEL assays were used to evaluate the cell apoptosis in liver tissues. (**D**), Masson staining assays were used to evaluate the fibrosis status in liver tissues. For all experiments, representative pictures are shown.

**Figure 7 ijms-24-05843-f007:**
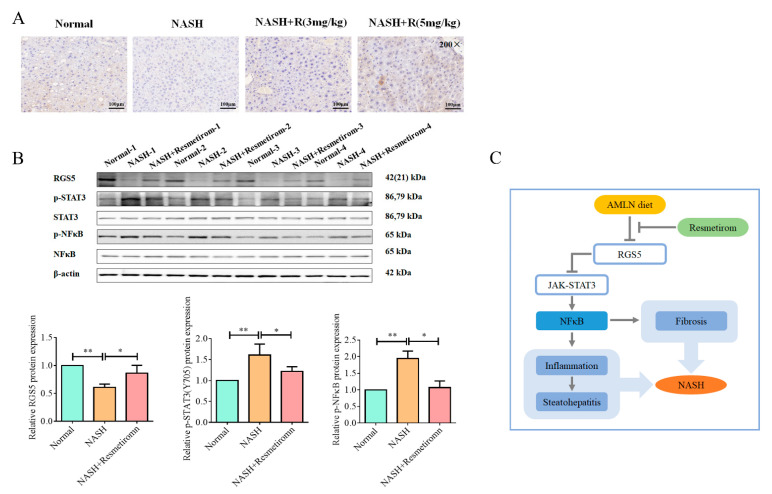
Resmetirom restored RGS5 expression and repressed the activation of STAT3 signaling pathway in NASH mouse model. (**A**), Sections of livers from mice in Figure 5 were used to perform the IHC assay to determine the RGS5 expression in liver tissue. Representative data are shown. (**B**), Total protein was extracted from livers in Figure 5, Western blot assay was performed to detect the expression of the indicated proteins. The protein level was analyzed and is presented in the histogram in the lower panel. (**C**), The potential underlying mechanism of how resmetirom improves NASH in a mouse model. Data are presented by mean ± SEM for three independent experiments. * *p* < 0.05; ** *p* < 0.01 vs. control.

**Table 1 ijms-24-05843-t001:** Primer sequences for qPCR assay.

Gene	Forward (5′-3′)	Reverse (5′-3′)
β-Actin	AGCAGTTGTAGCTACCCGCCCA	GGCGGGCACGTTGAAGGTCT
Col1a1	AAGAACCCTGCCCGCACAT	AGCCCTCGCTTCCGTACTCG
Ctgf	ATCTCCACCCGAGTTACCA	CGCAGAACTTAGCCCTGTATG
Spp1	TGATGAGACCGTCACTGCT	GCTGCCCTTTCCGTTGTTG
Lgals3	GGGAAAGGAAGAAAGACAGT	TTAGATCATGGCGTGGTTAG
Cd68	GATCTTGCTAGGACCGCTTAT	TGGTGGCAGGGTTATGAGT
Tnf	GGCGGTGCCTATGTCTCA	CCTCCACTTGGTGGTTTG
Il-1b	AGCATCCAGCTTCAAATC	ATCTCGGAGCCTGTAGTG
Ccl2	GCCTGCTGTTCACAGTTGC	TGGACCCATTCCTTCTTGG
Csf1	CACTGGGCACTAACTGGGTC	GCTCCTGGTGGTCTTCACG
CHAC1	CCTCCAGAGTTTACTGCCATGAC	GTAGGATCTCCGCCACTGATTC
SLC7A11	GCAGTTGCTGGGCTGATTTA	TGTTCTGGTTATTTTCTCCGAC
COL4A3	TCCTCACGGCTGGATTTCTC	GCACACCTGACAGCGACTT
RGS5	GAAACCAGCCAAGACCCAGAA	AGAGACCAACCTCTTTAGGAGCC
EGR3	TGCCTGACAATCTGTACCCC	TCCCAAGTAGGTCACGGTCT

## Data Availability

The data presented in this study are available on request from the corresponding author.

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
