# Peer review of "Resmetirom Ameliorates NASH-Model Mice by Suppressing STAT3 and NF-κB Signaling Pathways in an RGS5-Dependent Manner"

_ijms, 2023, doi:10.3390/ijms24065843_

Round 1

Reviewer 1 Report

Resmetirom, a THR beta agonist, is being investigated as a potential NASH treatment, but its mechanism of action is not well understood. Wang et al. used cultured cells and animal models to elucidate the mechanism of action of this drug. In cultured cell and animal models of NASH, administration of Resmetirom improved the pathophysiology of NASH. The mechanism of action of Resmetirom was shown to be suppression of liver inflammation by inhibiting STAT-3 and its downstream NF-kB activation through RG5 expression.

Comments

Although the experiments were performed in a very straightforward manner and the interpretation of the results is reasonable, several questions remain.

1.     It is clear that AMLN mice treated with resmetirom, whether at 3 mg/kg or 5 mg/kg, show improvement in lipid accumulation after 48 days of treatment, and consistent with the result of human trial. On the other hand, about improvement in fibrosis, authors demonstrated decreased expression of relevant genes, but not pathological results. It would be good to show MT staining (like shown in Figure 4).

2.     In this model, weight loss is seen after resmetirom administration. This paper shows that resmetirom acts specifically on RG5, but can you prove that the action of resmtirom is independent of weight loss?

3.     The expression of THR beta is not altered in cultured cells or animal models?

Author Response

Reviewer 1

Resmetirom, a THR beta agonist, is being investigated as a potential NASH treatment, but its mechanism of action is not well understood. Wang et al. used cultured cells and animal models to elucidate the mechanism of action of this drug. In cultured cell and animal models of NASH, administration of Resmetirom improved the pathophysiology of NASH. The mechanism of action of Resmetirom was shown to be suppression of liver inflammation by inhibiting STAT-3 and its downstream NF-kB activation through RG5 expression.

Comments

Although the experiments were performed in a very straight forward manner and the interpretation of the results is reasonable, several questions remain.

  1. It is clear that AMLN mice treated with resmetirom, whether at 3 mg/kg or 5 mg/kg, show improvement in lipid accumulation after 48 days of treatment, and consistent with the result of human trial. On the other hand, about improvement in fibrosis, authors demonstrated decreased expression of relevant genes, but not pathological results. It would be good to show MT staining (like shown in Figure 4).

Response: Thank a lot for this good suggestion. Masson staining assay was performed in liver tissues from groups of control, NASH, and NASH+ resmetirom (3 mg/kg or 5 mg/kg). The representative pictures are shown in Figure. 6D.

  1. In this model, weight loss is seen after resmetirom administration. This paper shows that resmetirom acts specifically on RG5, but can you prove that the action of resmtirom is independent of weight loss?

Response: Thanks a lot for this good question. We believe the underlying mechanism for the roles of resmetirom in NASH is very complicated. Our data just provides a novel insight into how resmetirom improves NASH, which is via RGS5 at least partially. At present, we cannot prove the function of resmtirom is independent of weight loss. Mice were fed with high-fat diet in the processes of NASH model establishment and drug treatment. It is reasonable to consider that the body weight difference  (but not loss) in two groups is resulted from the role of resmetirom on lipid metabolism.

  1. The expression of THR beta is not altered in cultured cells or animal models?

Response: Thanks a lot for this question. It is well known that resmetirom is a highly selective thyroid receptor β agonist. In this study, we did not examine the expression of THR beta in cells or liver tissues. As we do not keep this antibody and its delivery time is more than two weeks, we can not detect this protein by WB assay within the given time for revision. We will finish this experiment in our future investigation.

Reviewer 2 Report

Dear authors,

I have read with interest your paper. It is well-written and the concept of the study is fairly constructed and conducted during the trial.

In my opinion, it should be emphasized in the title that you will study NASH-model mice, because one could expect to read a human-based study. A title like ”...NASH mice model by suppressing...” could take it to the maximum fairness level. 

Another thing I find disturbing is that 3 of the 5 keywords do not find a place in the Introduction chapter. I find this unacceptable, they have to be briefly explained, and some relevant references should cite their description. 

I find the Materials and methods section written with enough details in order for one to be able to replicate your experiment - this is a good thing. Maybe, a more detailed explanation of why you have used the TUNEL assay could be useful.

The phrase from lines 264-265 should be rewritten. 

I think some of the explanations offered in the results section could be added and discussed directly in the Discussions section, thus enriching that important part of the article. The whole section comprises only 10 newly cited references (from 18-27), which I find maybe not be enough for a study that had so many aims and sub-studies. 

Also, as you have highlighted the strength of your study, you should also write a phrase that acknowledges the limitations of the present study.

All in all, I think the paper is fair and well-written, so these changes I will consider these as ”Minor revisions”. Good luck!

Author Response

Reviewer 2

I have read with interest your paper. It is well-written and the concept of the study is fairly constructed and conducted during the trial.

In my opinion, it should be emphasized in the title that you will study NASH-model mice, because one could expect to read a human-based study. A title like ”...NASH mice model by suppressing...” could take it to the maximum fairness level.

Response: Thanks a lot for this good advice. The title has been improved.

Another thing I find disturbing is that 3 of the 5 keywords do not find a place in the Introduction chapter. I find this unacceptable; they have to be briefly explained, and some relevant references should cite their description.

Response: Thanks a lot for this good suggestion. We have added more information for RGS5, STAT3, NF-κB with several references cited in the section of introduction.

I find the Materials and methods section written with enough details in order for one to be able to replicate your experiment - this is a good thing. Maybe, a more detailed explanation of why you have used the TUNEL assay could be useful.

Response: Thanks a lot for this question. The purpose for TUNEL assay is to examine the cell death statue in livers of mice from different groups. We also added this information in the section of TUNEL assay.

The phrase from lines 264-265 should be rewritten.

Response: Thanks a lot for this advice. We have rewritten the sentence.

I think some of the explanations offered in the results section could be added and discussed directly in the Discussions section, thus enriching that important part of the article. The whole section comprises only 10 newly cited references (from 18-27), which I find maybe not be enough for a study that had so many aims and sub-studies.

Also, as you have highlighted the strength of your study, you should also write a phrase that acknowledges the limitations of the present study.

Response: Thanks a lot for this good suggestion. We have improved the section of discussion. Especially more references have been cited.

Reviewer 3 Report

This manuscript investigated the protective role of Resmetirom in NASH and its underlying mechanism. They found that resmetirom alleviates NASH by decreasing lipid accumulation, fibrosis, and inflammation. The favorable effect of resmetirom on NASH is dependent on RGS5 expression and STAT3 and NF-kB signaling pathways. Overall, the manuscript is straightforward. However, there are some major concerns with this study.

1, Figure 1,2,4,5, and 6. The functional effect of resmetirom in NASH is well studied in the past, which dampens the novelty of this study.

2, The relationship of resmetirom and RGS5 seems to be the only highlight of this study. However, the functional role of RGS5 is not well studied in vivo.

Author Response

Reviewer 3

This manuscript investigated the protective role of Resmetirom in NASH and its underlying mechanism. They found that resmetirom alleviates NASH by decreasing lipid accumulation, fibrosis, and inflammation. The favorable effect of resmetirom on NASH is dependent on RGS5 expression and STAT3 and NF-kB signaling pathways. Overall, the manuscript is straightforward. However, there are some major concerns with this study.

1, Figure 1,2,4,5, and 6. The functional effect of resmetirom in NASH is well studied in the past, which dampens the novelty of this study.

Response: Thanks a lot for this question. According to the publishes in Pubmed, our study is probably the first report in which the role of resmetirom in NASH was investigated at cellular level, and high throughput sequencing was used to illustrate the underlying mechanism. Our work provided a novel insight into how resmetirom acts in NASH.

2, The relationship of resmetirom and RGS5 seems to be the only highlight of this study. However, the functional role of RGS5 is not well studied in vivo.

Response: Thanks a lot for this good question. Although resmetirom has been proven to exert amazing potentials in NASH treatment in animal experiments and clinical trials, the underlying mechanism is still poorly documented. Our data might be the first research giving a preliminary finding about it. In the present study, RGS5 was identified to take part in the functions of resmetirom. And we only detected the RGSs expression in liver tissues in NASH mice or resmetirom treated mice. Indeed, RGS5 knockout transgenic mouse should be applicated to further confirm the role of RGS5 on the functions of resmetirom in NASH treatment. This experiment will be performed in our future investigation. We have added the statement that acknowledges the limitations of this study in the section of discussion.

Round 2

Reviewer 1 Report

Thank you for adding the diagram. Unfortunately, this figure does not show that resmestirom has a very impressive effect on fibrosis. As shown in the phase before resmetirom administration (Figure 4E), it would be better to present a strong fibrosis picture as a control. However, it could be interpreted as if the short administration (48 days) did not show a significant difference in fibrosis histologically.

Author Response

Response: Thanks a lot for this question. We re-performed the Masson staining assay and used another machine to scan the sections. This time, we believe the result is much better. The new data was shown in fig.6.

Round 3

Reviewer 1 Report

Thank you for your work.